# Effects of sensorimotor delays and muscle force capacity limits on the performance of feedforward and feedback control in animals of different sizes

Sayed Naseel Mohamed Thangal[1], Heather L. More[1], C. David Remy[2], J. Maxwell Donelan[1]*

**1** Department of Biomedical Physiology and Kinesiology, Simon Fraser University, Burnaby, British Columbia, Canada, **2** Institute for Adaptive Mechanical Systems, University of Stuttgart, Stuttgart, Baden-Württemberg, Germany

* mdonelan@sfu.ca

## Abstract

Animals rely on feedforward and feedback control for perturbation responses. When comparing terrestrial mammals of different sizes, we generally find that several features that affect perturbation responses change—larger animals have longer sensorimotor time delays, heavier body segments, and proportionally weaker muscles. We used simple computational models to study how control of fast perturbation responses is constrained by two limitations—sensorimotor delays and muscle force capacity—as a function of animal size. We developed two tasks representing common perturbation response scenarios in animal locomotion: a distributed mass pendulum approximating swing limb repositioning (swing task), and an inverted pendulum approximating whole body posture recovery (posture task). First, we used a normalized feedback control system to show how feedback response times can either be limited by the force generation capacity of muscles (force-limited), or by sensorimotor delays which constrain the maximum feedback gains that can be used to produce stable responses (delay-limited). Next, we used more detailed scaled models which represent the full size range of terrestrial mammals and parameterized the sensorimotor delays, maximum muscle forces, and inertial properties using scaling relationships from literature. Across animal size and in both tasks, we found that feedback control was primarily delay-limited—the fastest responses used a fraction of the available muscle force capacity. We compared feedback control to the fastest feedforward control strategy, and found that feedforward control response times were about four times faster than feedback control in the smallest animals, and around two times faster in the largest animals. For rapid perturbation responses, feedback control appears ineffective for terrestrial mammals of all sizes, as our simulated fastest response times exceeded available movement times. Thus, feedforward control strategies—including anticipatory adjustments, ballistic motor programs, and exploitation

**Data availability statement:** All relevant data are within the manuscript and its Supporting information files. We have shared the codes used to generate our results through a Github repository. https://github.com/sayednaseel/GITHUB-ScalingOfFeedbackControl.

**Funding:** This work was supported by the Natural Sciences and Engineering Research Council of Canada with a Discovery Grant no: RGPIN326825 (J.M.D) and by Simon Fraser University with a Multi Year Funding-Internal award (S.N.M.T). The funders had no role in study design, data collection and analysis, decision to publish, or preparation of the manuscript.

**Competing interests:** The authors have declared that no competing interests exist.

of intrinsic musculoskeletal dynamics—may be essential for reacting quickly to sudden and large perturbations in terrestrial mammals.

## Author summary

All animals need to respond quickly when perturbed. But many physiological features that affect how quickly an animal can respond—such as the length of nerves, the weight of body segments, and the strength of muscles—depend on animal size. Here we use computational models to study how perturbation responses in animals of different sizes are constrained by two limitations—sensorimotor time delays and muscle force capacity. In simulations that span the size range from shrews to elephants, we find that feedback control is predominantly limited by time delays, constraining animals to using only a fraction of their available muscle force when responding to a perturbation. These time delays make our feedback control implementation consistently too slow for rapid perturbation response tasks. We compared feedback control to feedforward control, and found that feedforward control avoids some of the limitations of time delays and can use all available muscle force. It was about four times faster than feedback control in the smallest animals, and around two times faster in the largest animals. This supports the view that feedforward control strategies are essential for reacting quickly to sudden and large perturbations in terrestrial mammals.

## 1. Introduction

Animals rely on both feedforward and feedback control strategies to respond to perturbations they encounter when moving through the world. Consider an animal whose foot gets caught on a vine, requiring a quick perturbation response to prevent a fall and injury. Sensors (cutaneous, proprioceptive, vestibular etc.) would detect the perturbation and transmit the signal through afferent nerves to neural controllers in the spinal cord and brain. These neural controllers could then trigger a stumble corrective response or elevation strategy that involves lifting and swinging the foot forward to a new position or "reference target" [1,2]. The neural controllers could use either feedforward control or feedback control to determine the muscle activation (motor) commands to perform the perturbation response. Feedback control takes in sensory information about the body's movements and compares it to the desired reference target throughout the response. The controller uses this changing error signal to generate the motor commands that accurately reposition the body to the new stable position [3,4]. In contrast, feedforward control uses predefined motor commands that reposition body segments to the reference target without continuously updating its response based on sensory feedback [5–7]. One benefit of this strategy is that it can be used to rapidly reject a perturbation as it does not have to wait for continuous adjustment by sensory feedback.

In reality, animals use neural controllers that simultaneously leverage feedforward and feedback control, perhaps to benefit from their individual advantages while compensating for each other's drawbacks [7–9]. Computational models developed to study animal movement often implement a control system with simultaneous feedforward and feedback strategies [10–12]. Once a perturbation occurs, neural controllers are triggered which determine a reference target that the body (plant) has to be repositioned to in order to regain stability. A feedforward controller then uses predetermined motor commands to reposition body segments to the reference target, without using sensory feedback. A feedback controller works in parallel, generating motor commands based on the error between the same reference target and sensory feedback, addressing any deviations which were not predicted by the feedforward controller. Neural controllers might prioritize feedforward or feedback control depending on factors such as the body segments being moved, time available and sensory information [7,10]. Anticipatory postural adjustments provide clear examples of feedforward control; humans activate specific muscle groups prior to predictable perturbations to maintain balance [13]. Fast, goal-directed ballistic movements, such as saccades and rapid reaching, also reflect feedforward-dominated control, where preplanned motor commands execute without time for feedback corrections [14]. Feedback control, however, remains fundamental for stability and adaptability, especially when perturbations are unpredictable or evolve over longer timescales. The classic servo-control model of postural balance by Nashner and McCollum (1985) demonstrated how delayed sensory feedback, integrating vestibular, proprioceptive, and visual inputs, could stabilize upright posture through adaptive gains [15]. Empirical studies by Horak and colleagues (1996) further showed that human postural responses dynamically reweight sensory feedback sources depending on context and perturbation characteristics [16]. Kiemel and colleagues conducted experiments to perform system identification of the musculoskeletal and feedback components involved in the neural control of upright posture, and their results are consistent with a feedback controller that aims to reduce body sway while using minimal muscle activation. [17,18]. In locomotion, reflex-mediated feedback loops—such as stretch reflexes and load-sensitive responses—are essential for adjusting muscle activity to terrain changes and unexpected disturbances, as seen in walking cats [19] and human stumbling recovery [1]. Moreover, Ting and colleagues have shown that time-delayed feedback models can reproduce complex EMG activation patterns during postural perturbations, underscoring feedback's role in shaping corrective responses even when delayed [20–23]. Together, these examples illustrate that animals rely on control strategies that range from fast, predictive feedforward mechanisms to slower but adaptable feedback responses.

The performance of feedback and feedforward control systems is limited by time delays and actuator force capacity. The limited speed at which signals are conducted along nerves and across synapses results in sensorimotor time delays [24]. These time delays are not just present in biological control systems—electrical signal transmission and data processing time delays constrain the performance of engineering control systems [25]. These time delays increase the time it takes to initiate a response in both feedforward and feedback systems. Additionally, there is a limit to the maximum forces that animal muscles can produce [26], which parallels the force saturation limits seen in engineered actuators [25]. Time delays and actuator force capacity are two well studied limitations in engineering control systems [27–29]. Actuator force capacity can reduce the responsiveness of both feedforward and feedback control systems because the speed at which a physical system can be repositioned depends on how quickly the system can be accelerated, and this acceleration depends on the force capacity of its actuators. The fastest neurally mediated control strategy to reject a perturbation is equivalent to feedforward control with a minimum-time implementation [30,31]. It is an open-loop, pre-planned motor command without using sensory feedback [32–34]. As such, it represents the theoretical upper limit of response speed achievable by a control strategy, constrained only by initial sensorimotor delay, actuator force capacity, and system dynamics.

Feedback systems have an additional vulnerability to time delays. When feedback control signals are outdated due to time delays, the motor commands generated are inappropriate for the current state of the system, resulting in poor control and reduced stability [3,35]. If the delays grow too large, the delayed feedback signal will actively destabilize the system instead of controlling it. To compensate for long feedback delays, a feedback-delayed controller often requires lower gains to remain stable when compared to a controller with shorter time delays. This results in lower forces and slower

responses. The main limiting factor in a feedback control system could either be its time delays or its actuator force capacity. A feedback system that doesn't suffer from time delays will rapidly reach the force capacity of its actuators and how quickly it can respond will then depend upon how strong the actuators are at full capacity. In contrast, a feedback system that has long time delays, will never be able to tap into the full force capacity of its actuators because the long delays will necessitate low feedback gains to insure stability. Whether a feedback control system is force-limited or delay-limited will depend on the relative magnitudes of actuator strength and time delay duration.

We suspect that the effects of time delays and actuator force capacity on control performance depends on animal size. Sensorimotor delays are longer in larger animals when compared to smaller animals, even when expressed relative to movement time [24]. Considering cursorial terrestrial mammals, larger animals generally have larger and stronger muscles when compared to smaller animals, but their body segments are also heavier rendering their muscles proportionally weaker [24,36–40]. However, larger animals also benefit from having more time available to correct disturbances. For example, stride frequency decreases with increasing body size, as shown by Heglund and Taylor (1988) across a broad range of terrestrial mammals [41], leading to longer stride and swing durations [24,42]. Since swing duration determines the time window to reposition a limb after a trip, larger animals have more time to complete such corrective responses during locomotion. Similarly, when considering postural perturbations, larger animals take longer to fall to the ground due to their longer limbs and greater height [38], providing more time for corrective actions. Because these biomechanical features scale with animal size, whether feedback control is limited by time delays or force capacity may depend on animal size. Consequently, the ability to control movement using feedforward and feedback strategies—and the relative advantage of each—may also depend on animal size.

Here our objective was to understand the performance of feedforward and feedback control strategies in animals of different sizes when constrained by sensorimotor delays and muscle force capacity limits. To accomplish this, we developed simple and separate feedforward and feedback control systems and scaled them to represent the size range of terrestrial mammals. Importantly, our feedforward controller was designed as a time-optimal, open-loop benchmark—an idealized representation of the fastest possible response achievable. This abstraction serves to define an upper bound on feedforward performance, distinct from passive feedforward strategies that exploit intrinsic musculoskeletal dynamics. The consequences of time delays and actuator force capacity are most evident when a system is required to respond as fast as possible, because fast responses require using the maximum possible muscle forces while ensuring that the response remains accurate and stable. Two fast response situations in animals are repositioning the swing leg after a stumble when running at high speed and counteracting an aggressive push when standing still. Consequently, we quantified response times for two perturbation response tasks: a forelimb swing leg repositioning task under the control of shoulder muscles (swing task), and a posture recovery task under the control of distal muscles of all four legs (posture task). We parameterized the models' inertial, muscular and neural features based on animal scaling data from the literature and optimized their controller parameters to achieve the fastest responses. We simulated perturbation responses in these parameterized models to estimate the scaling relationship between response time and animal size for both feedforward and feedback control. By comparing response times under these two different types of control to each other, and to available movement times, we quantified the effectiveness of these control strategies across animal size and determined whether response times are limited by time delays or force capacity.

## 2. Methods and models

To understand fast perturbation responses in animals, we developed two types of models with different levels of complexity (scaled models and a normalized feedback control model), considered two limitations to control performance (sensorimotor delays and muscle force capacity limits), and simulated two perturbation response scenarios (swing task and posture task). The scaled models are more detailed, consider gravity, separately simulate feedforward and feedback control, and are parametrized to represent the size range of terrestrial mammals. The normalized feedback model is a

simplified version of the scaled feedback models that has only a single free parameter, allowing us to more clearly evaluate how sensorimotor delays and muscle force capacity limits interact to affect perturbation responses.

To improve clarity and avoid repetition, we present subsections in different orders within the Methods and Results sections. In this section, we first describe our more elaborate scaled models (2.1) and then list the simplifications made to develop the normalized feedback model (2.2). In the subsequent Results section, we first describe predictions from the normalized model (3.1), and then validate those predictions on our scaled models (3.2).

## 2.1 Scaled models

We developed scaled models that can represent the size range of terrestrial mammals using pendulums to represent the body segments, and parametrized them using data for inertial, muscular and neural features from the scaling literature. Fig 1 provides a schematic overview of the feedforward and feedback control system models.

**Feedback control**

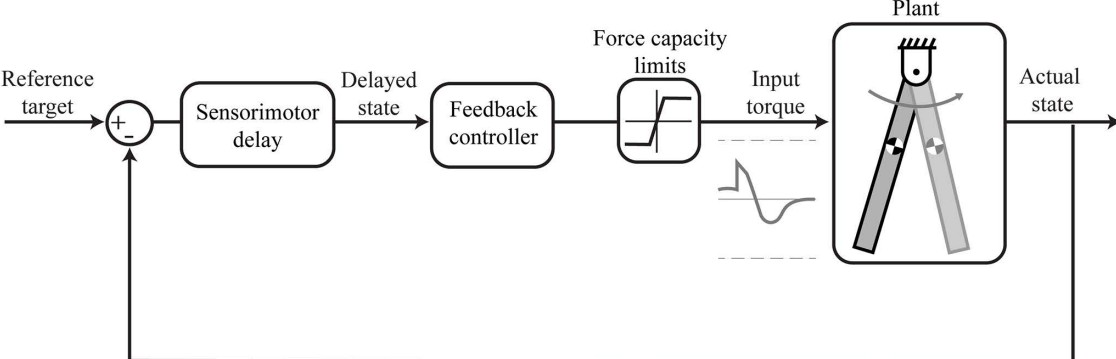

**Feedforward control**

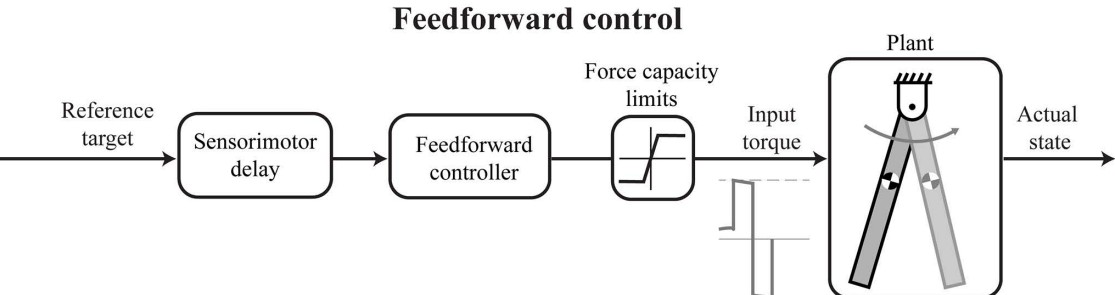

**Fig 1. Feedback and feedforward control systems for the scaled models.** The perturbation response is triggered by an external event such as a trip or a push to the body. The body segments are represented here by the *plant* using rigid body dynamics and pendular models. The *sensorimotor delay* block represents all the time delays in the neural control pathways from detection of the perturbation, transmission through sensory and motor nerves, to synaptic computations and activating muscles. Once the neural controllers receive sensory information about the perturbation, they determine a perturbation response that involves repositioning the body to a desired *reference target*. The feedback system (top) uses a proportional-derivative (PD) controller to generate torque based on time-delayed state feedback. The feedforward system (bottom) optimizes controller torques to minimize response time resulting in bang-bang actuation, serving as an upper bound benchmark for the fastest possible perturbation response. The *force capacity limits* block restricts the maximum positive and negative torques that can be applied by the controller, based on the force capacity of the muscles used in the task. We have shown representative torque profiles for the feedback and feedforward systems beside the plant (not to scale).

**2.1.1 Swing task. Plant dynamics**. The swing task represents a forelimb trip during early swing, requiring the animal to use its shoulder muscles to reposition its leg to prevent a fall. This is similar to the stumble corrective response in animals [2,43] and the elevating strategy in humans [1]. We modeled the forelimb as a distributed mass pendulum and the task required repositioning the limb under muscle flexor and extensor torques applied at the shoulder, from rest at an initial negative angle to rest at a final positive angle (Fig 3a). The equations of motion are given by:

$$I_{limb}\,\ddot{\theta}(t) = \tau_{total} = \tau_{act} + \tau_{gravity} \tag{1}$$

where $\tau_{act}$ represents the *actual torque* exerted by the shoulder muscles. This torque can be negative-valued or positive-valued where the sign indicates whether the extensor muscle or the antagonist flexor muscle is active. $\tau_{gravity}$ represents *gravitational torque*. $I_{limb}$ represents the *moment of inertia* of the forelimb about the shoulder. $\ddot{\theta}$ represents the *angular acceleration* of the forelimb. We adopted the convention that the angle is zero when the limb is pointing vertically downwards and defined the counterclockwise direction to be positive. The torque due to gravity is described by:

$$\tau_{gravity}(t) = -M_{limb}\,g\,L_{COM}\,\sin\theta(t) \tag{2}$$

where $M_{limb}$ is the *mass of the forelimb*, and $L_{COM}$ is the *distance from the shoulder joint to the limb center of mass*. $g$ is the *acceleration due to gravity* (9.81 m/s$^2$).

**Feedback controller**. Under feedback control, $\tau_{act}$ represents the *actual torque* applied at the shoulder, and $\tau_{des}$ represents the *desired torque* specified by the Proportional-Derivative (PD) controller output. $\tau_{act}$ and $\tau_{des}$ are not necessarily equal as $\tau_{des}$ is subject to force capacity limits:

$$\tau_{act} = sat\left(\tau_{des}\right) = \begin{cases} \tau_{iso} & \text{if } \tau_{des} > \tau_{iso} \\ \tau_{des} & \text{if } -\tau_{iso} \le \tau_{des} \le \tau_{iso} \\ -\tau_{iso} & \text{if } \tau_{des} < -\tau_{iso} \end{cases} \tag{3}$$

$\tau_{iso}$ represents the force capacity limit, set as the *maximum isometric torque* that can be applied by muscles about the shoulder joint to reposition the swing limb. In this feedback controller, $\tau_{des}$ is defined as:

$$\tau_{des}(t) = \begin{cases} 0 & \text{if } 0 < t < t_{SM} \\ K_p\left[\theta_r - \theta\left(t - t_{SM}\right)\right] + K_d\left[-\dot{\theta}\left(t - t_{SM}\right)\right] + \tau_{steadystate} & \text{if } t \ge t_{SM} \end{cases} \tag{4}$$

where $K_p$ represents the *proportional gain*, $K_d$ represents the *derivative gain*, $\theta_r$ represents the *reference target* (the final desired position of the limb), $\tau_{steadystate}$ represents the *steady state torque* required to counter gravity at the final state, and $t_{SM}$ represents the *sensorimotor time delay*. The sensorimotor delay represents all the time delays in the neural control pathways including the detection of the perturbation by sensors (proprioceptive, cutaneous, vestibular), transmitting the signals through the sensory and motor nerves, computing perturbation responses at spinal synapses, and activating and generating muscle forces [24]. We implemented an initial deadtime at the start of the perturbation response equal to the sensorimotor delay, during which the controller does not apply any forces, and the forelimb moves under the influence of gravity. Once this deadtime is over, the controller begins to exert forces on the pendulum, using time-delayed state information.

**Feedforward controller**. Under feedforward control, the plant dynamics are identical, but the *actual torque* ($\tau_{act}$) is found by an optimization to minimize the time required to complete the movement; this results in bang-bang actuation—a

control strategy that switches instantaneously between our estimates of the maximum isometric flexion and extension torques [31–34]. Our bang-bang controller also has an initial deadtime equal to the sensorimotor time delay:

$$\tau_{act}(t) = \begin{cases} 0 & if\ 0 < t < t_{SM} \\ +\tau_{iso} & if\ t_{SM} < t < t_{switch} \\ -\tau_{iso} & if\ t_{switch} < t < t_{end} \end{cases}$$

(5)

where $t_{switch}$ represents the *switch time* at which the controller changes the direction of applied torque. The feedfoward controller does not use sensory information; we determine $t_{switch}$ by optimizing for fastest response time. Table 1 summarizes the parameters used in this swing task scaled model, as well as how we determined the values.

**Perturbations**. In feedforward control responses, the response time depends on the size of the movement—small movements are accomplished quickly, but large movements take more time. Response time is independent of movement size in a linear PD control system (without nonlinearities like gravity or saturation limits) if the controller gains are not changed—small and large movements take the same amount of time. Linear PD control systems decay exponentially, moving a certain percentage of the movement to the reference target in a given amount of time. As settling time is also defined as a percentage of the movement to the reference target, the response time becomes invariant to movement size [44]. Larger movements create larger error terms (Eqn 4), resulting in larger controller torques that reposition the body within the same amount of time. While our feedback control system does consider nonlinear behaviour (gravity, saturation limits, steady state torque, and initial deadtime), the linear PD control behaviour dominates. This results in feedback response times changing very little with movement size. Thus, to compare response times in the two types of control, it is useful to pick a movement size. Here we used a swing leg repositioning from -15° to +15° because sensorimotor delays and inertial delays are equally matched in a one kg animal for this movement size [37].

**Table 1. Table of input parameters, optimized parameters, and results for the swing task.** The Source column indicates whether the parameter was an input with values calculated from the scaling literature, or whether the parameter or performance measure was found through optimization. The Coefficient and Exponent columns are the coefficients and exponents in the power law equations that quantify the effect of size on that parameter.

| Parameter | Source | Coefficient ($a$) | Exponent ($b$) |
|---|---|---|---|
| **Swing task inputs** | | | |
| $t_{SM}$: Sensorimotor delay (ms) | Literature | 31 | 0.21 |
| $M_{limb}$: Forelimb mass (kg) | Literature | $5.8 \times 10^{-2}$ | 1.00 |
| $I_{limb}$: Forelimb inertia (kg.m²) | Literature | $2.52 \times 10^{-4}$ | 1.75 |
| $L_{COM}$: COM length (m) | Literature | $5.6 \times 10^{-2}$ | 0.36 |
| $\tau_{iso}$: Triceps torque (N.m) | Literature | 0.54 | 1.19 |
| **Swing task outputs** | | | |
| Swing duration at max sprint (ms) | Literature | 147.9 | 0.17 |
| **Feedback control results** | | | |
| $K_p$: Proportional gain (N.m/rad) | Optimization | $1.75 \times 10^{-2}$ | 1.28 |
| $K_d$: Derivative gain (N.m/(rad/s)) | Optimization | $4.76 \times 10^{-3}$ | 1.54 |
| Feedback response time (ms) | Optimization | 198.9 | 0.21 |
| **Feedforward control results** | | | |
| $t_{switch}$: Switch time (ms) | Optimization | 46.3 | 0.23 |
| Feedforward response time (ms) | Optimization | 61.9 | 0.24 |

**Simulations and optimization – feedback control**. For the feedback control simulations, which required time-delayed state information, we used the dde23 delay-differential equation solver (explicit Runge-Kutta algorithm with discontinuity tracking) to numerically integrate the equations of motion (MATLAB R2024b, The MathWorks, Inc., Natick, MA, USA) [45]. We calculated both settling time and overshoot on the angle curve (Fig 3b). We quantified response time as the settling time with 2% thresholds of the entire movement from the initial position to the *reference target* ($\theta_r$) (Fig 3b). We constrained overshoot of the angle curve to be zero, to match a critically damped step response [46]. We normalized the simulation time by simulating for 20 $t_{SM}$. To counter the steady state gravitational torques at the target position of +15°, we explored the use of a *steady state torque* ($\tau_{steadystate}$), or integral control. We used the steady state torque option because it required fewer parameters to optimize and produced faster response times. We used MATLAB optimization functions fminsearch (simplex algorithm) and fmincon (interior-point algorithm) to search for the ideal parameters for the feedback controller ($K_p$, $K_d$) that produced the fastest response times without overshoot.

**Simulations and optimization – feedforward control**. As the feedforward simulations did not require time-delayed state information, we used MATLAB's ode45 variable time step solver (explicit Runge-Kutta algorithm) to numerically integrate the equations of motion. Under feedforward control, we used event detection to terminate the simulation when the pendulum came to rest and used the elapsed time as response time. We used optimization to find the ideal parameter for the feedforward controller ($t_{switch}$) that stopped the forelimb at the reference target, while achieving the fastest response time.

**Scaling relationships**. Two recent publications that quantified the scaling of mammalian anatomical and neural features, and an older publication on the scaling of muscle features in mammals, provide us with sufficient data to parametrize our scaled control systems [24,36,38]. We used data from More and Donelan (2018) on the monosynaptic stretch reflex pathways to parametrize the sensorimotor delay ($t_{SM}$) [24]. The sensorimotor delays consider six component delays within the reflex pathways: sensing delay, nerve conduction delay, synaptic delays, neuromuscular junction delay, electromechanical delay and force generation delay. We used data from Kilbourne and Hoffman (2013) on terrestrial mammalian limbs to set inertial properties in this model ($M_{limb}$, $L_{COM}$, $I_{limb}$) [38]. We estimated the maximum force capacity of the shoulder flexor and extensor muscles ($\tau_{iso}$) using muscle mass, length, and moment arm data for the triceps published by Alexander et al. (1981) [36]. While one of the three heads of the triceps—the triceps brachii caput longum—does connect to the scapula and flex the shoulder joint [47], it is not this joint's main muscle. We nevertheless used Alexander's scaling data for the triceps to parameterize the shoulder flexors and extensors, as it is the only literature reference to our knowledge that provides the parameters necessary to estimate muscle torque at the shoulder. Bishop et al. (2021) found similar scaling relationships as those we used here for combined forelimb proximal muscle mass and length [48]. However, they did not provide values for muscle moment arms, so we elected to use Alexander's triceps values. We found muscle volume from mass by assuming a density of 1060 kg/m³ [49], and cross-sectional area by dividing muscle volume by muscle length. We did not consider muscle pennation and its effects on physiological cross-sectional area in our models. We then multiplied cross-sectional area by the isometric stress of muscles, estimated to be 20 N/cm² [26,50], to get muscle force. $\tau_{iso}$ is muscle force multiplied by the moment arm. We evaluated the models for eight logarithmically spaced animal sizes from 1 gram to 10 tonnes to cover the size range of terrestrial mammals, and also included 5 grams and 5 tonne sizes for 10 total animal sizes [51,52]. We then performed a least squares linear regression on the logarithmically transformed animal mass vs. simulation output data (controller gains and response times) to extract the coefficient and exponent of the underlying scaling relationship [53].

**2.1.2 Posture task. Plant dynamics**. The posture task represents an animal recovering its posture after a forward push to the body, under the control of the ankle plantarflexors [21,23,54–56]. We used a point mass inverted pendulum to represent the entire body, with a mass $M$ equal to the weight of the whole animal, and length $L$ set to the leg length for each animal mass (Fig 5a). We modeled forward push perturbations as impulsive forces that result in changes to the body's initial forward velocity, but not its position. The task required generating torques to counter the initial velocity

caused by the push and returning to rest at the vertical position from which the system started. The equations of motion are given by:

$$I\ddot{\theta}(t) = \tau_{total} = \tau_{act} + \tau_{gravity}$$

(6)

where $\tau_{act}$ represents the *actual torque* exerted by the muscles. This torque can be positive-valued or negative-valued, where the sign indicates whether the plantarflexors or dorsiflexors are active. $\tau_{gravity}$ represents the *torque exerted by gravity*. *I* is the *moment of inertia* of the pendulum, computed as $ML^2$. In contrast to the swing task, we defined the angle to be zero when the inverted pendulum is pointing vertically upwards and defined the counterclockwise direction to be positive. The torque due to gravity is consequently defined as:

$$\tau_{gravity}(t) = MgL \sin \theta(t)$$

(7)

The feedback controller in the posture task did not require a steady state term, as there are no gravitational torques at the vertical end position. Other than this difference, we determined $\tau_{act}$ for feedforward and feedback control in the posture task using identical equations to the swing task (Eqns 3,4,5). Table 2 summarizes the parameters used in the posture task scaled models, as well as how we determined the values.

   **Perturbations**. As in the swing task, we had to choose a perturbation size to compare feedback and feedforward response times. To trigger postural perturbations of similar magnitudes in different sized animals, we scaled the initial linear velocity for the inverted pendulum mass with a dimensionless velocity of 0.21 Froude number [57]. At this perturbation size, the sensorimotor delays equaled the inertial delays in a one kg animal [37]. We defined the target angle and velocity to be zero, to bring the pendulum to rest at the vertical position.

   **Simulations and optimization-feedback control**. Under feedback control, as the posture task involved a velocity perturbation, we calculated settling time with 2% thresholds on the angular velocity curve (Fig 5b). As overshooting the vertical position during a posture correction can also result in loss of balance, we calculated overshoot on the angle curve and constrained it to be zero during the optimization.

**Table 2. Table of input parameters, optimized parameters, and results for the posture task.**

| Parameter | Source | Coefficient (*a*) | Exponent (*b*) |
|---|---|---|---|
| **Posture task inputs** | | | |
| $t_{SM}$: Sensorimotor delay (ms) | Literature | 31 | 0.21 |
| *I*: Whole body inertia (kg.m²) | Literature | $2.64 \times 10^{-2}$ | 1.74 |
| *L*: Center of mass height (m) | Literature | $1.62 \times 10^{-1}$ | 0.37 |
| $\tau_{iso}$: Plantarflexor torque (N.m) | Literature | 3.41 | 1.21 |
| **Posture task outputs** | | | |
| Time to fall leg length (ms) | Literature | 182.0 | 0.19 |
| **Feedback control results** | | | |
| $K_p$: Proportional gain (N.m/rad) | Optimization | 5.64 | 1.34 |
| $K_d$: Derivative gain (N.m/(rad/s)) | Optimization | $5.80 \times 10^{-1}$ | 1.53 |
| Feedback response time (ms) | Optimization | 239.2 | 0.22 |
| **Feedforward control results** | | | |
| $t_{switch}$: Switch time (ms) | Optimization | 72.3 | 0.28 |
| Feedforward response time (ms) | Optimization | 94.6 | 0.28 |

**Simulations and optimization-feedforward control**. Under feedforward control, we used event detection to terminate the simulation when the inverted pendulum came to rest, and optimized $t_{switch}$ to stop the pendulum at the vertical position. Apart from these differences, we simulated and optimized using identical methods to the swing task.

**Scaling relationships**. We used the average length of the mammalian forelimb and hindlimb based on data from Kilbourne and Hoffman (2013) to set inverted pendulum length [38]. We set $\tau_{iso}$ to equal four times the maximum torque that can be exerted by a single set of ankle plantarflexors based on Alexander et al. (1981) [36].

### 2.1.3 Assumptions and approximations.

We made several assumptions to simplify the scaled models. For example, we assumed that the perturbation responses are mediated through monosynaptic spinal reflex pathways, and do not involve higher level (supraspinal or cortical) neural centers. While the term "feedforward control" is also used in the literature to refer to several different control strategies, including rhythmic signals generated by central pattern generators [9,58], we used a minimum-time feedforward controller with bang-bang actuation. We also simplified the scaled feedback models by assuming that all sensorimotor delays occur in the feedback (sensory) pathway. In real life, there are delays in both the feedback (sensory) pathway and the feedforward (motor) pathway. In S1 Text, we show how this assumption does not affect outcomes for the step response inputs we use in our simulations (section S1 of S1 Text). While we have assumed that the feedforward and feedback control systems experience the same total sensorimotor delays, Cao et al. (2025) show that the neural controllers might face different computational delay times when using feedforward vs. feedback control [11]. As we use simple controllers, both the feedforward controller (Eqn 5) and the feedback controller (Eqn 4) apply instantaneous torques after the initial deadtime. We did not model Hill-type muscles due, in part, to a lack of data to reasonably parametrize such models across the size range of terrestrial mammals. Instead, our sensorimotor time delay parameter includes two components—electromechanical delay and force generation delay—which roughly approximate how muscle takes time to generate force and how it scales across body size [24]. The electromechanical delay approximates the lag in force generation caused by muscle activation-deactivation dynamics. The force generation delay approximates the time required for muscle fibers to reach peak twitch force.

We tested the sensitivity of our results to certain parameters. Our choice of 2% for settling time threshold, while arbitrary, is conventional in control theory and allows us to be consistent in determining response times across animal size. A tighter tolerance of 1% resulted in the coefficient in the scaling of swing task feedback response times for a 30° movement increasing by 75%, while a looser tolerance of 4% resulted in the response time coefficient decreasing by 8%; the exponent remained largely unchanged. To check sensitivity to movement size, we evaluated response times in the swing task for the extreme case of a 180° movement size. The feedback response time coefficient reduced by 6%, while the exponent did not change. This matches predictions that PD control response times are not sensitive to movement size (Methods section 2.1.1), with small deviations due to the effects of steady state torque, initial deadtime and gravity. In contrast, the swing task feedforward response time coefficient increased by 66.4%, while the exponent increased by 3.7%, for the 180° repositioning. We used an exponent of 0.41 for the swing task moment arm based on data for the triceps muscle from Alexander et al. (1981), due to a lack of scaling information for shoulder muscles in the literature. However, the cited moment arm data is for the triceps muscle about the elbow, not the shoulder. Biewener (1990) reported that antigravity muscle moment arms for forelimb muscles scale with 0.44 [39]. We varied muscle moment arm exponents from 0.38 to 0.44 while keeping movement size at 30° and studied its effects on response times. Feedback response times did not change, as the feedback controllers were not able to reach the muscle force capacity limits in any case. The coefficient of the scaling of feedforward response times changed by 0.65% to -0.53% for moment arm exponents of 0.38 to 0.44 respectively. The exponent of the scaling of feedforward response times changed by +3% to -3% for moment arm exponents of 0.38 to 0.44 respectively. These results suggest that our conclusions are not very sensitive to the exponent we use for shoulder muscle moment arm.

## 2.2 Normalized feedback control system

In addition to the family of scaled feedback models we describe above, we developed a single normalized model to study how feedback control can be limited by time delay and force capacity. This normalized feedback model is a simplified version of the scaled feedback models and only has a single free parameter, allowing us to more clearly evaluate how sensorimotor delays and muscle force capacity limits interact to affect perturbation responses. This single model applies to both swing and posture tasks in animals of all sizes. To accomplish this, we simplified the scaled feedback models by ignoring gravity, thereby reducing the plant to a double integrator under feedback control. Before normalization, the *desired torque* $\tau_{des}$ produced by the feedback controller is given by:

$$\tau_{des}(t) = \begin{cases} 0 & \text{if } 0 < t < t_{SM} \\ K_p \left[\theta_r - \theta\left(t - t_{SM}\right)\right] + K_d \left[-\dot{\theta}\left(t - t_{SM}\right)\right] & \text{if } t \geq t_{SM} \end{cases} \tag{8}$$

To non-dimensionalize Eqn 8 (same as Eqn 4 without the steady state torque component), we used three characteristic parameters of the perturbation response: $I$ is the *moment of inertia* of the body segments being repositioned, $\theta_r$ is the *reference target* for feedback control and represents the size of the perturbation in the swing task, and $t_{SM}$ is the *sensorimotor time delay*. We divided each torque component in Eqn 8 by $\frac{I\theta_r}{t_{SM}^2}$, where $I$ has fundamental units of $M^1L^2$, $t_{SM}$ has units of T, and $\theta_r$ is dimensionless:

$$\overline{\tau}_{des} = \begin{cases} 0 & \text{if } 0 < \overline{t} < 1 \\ \overline{K}_p \left[1 - \overline{\theta}\left(\overline{t} - 1\right)\right] + \overline{K}_d \left[-\dot{\overline{\theta}}\left(\overline{t} - 1\right)\right] & \text{if } \overline{t} \geq 1 \end{cases} \tag{9}$$

$$\ddot{\overline{\theta}} = \overline{\tau}_{act} = sat\left(\overline{\tau}_{des}\right) = \begin{cases} \overline{\tau}_{iso} & \text{if } \overline{\tau}_{des} > \overline{\tau}_{iso} \\ \overline{\tau}_{des} & \text{if } -\overline{\tau}_{iso} \leq \overline{\tau}_{des} \leq \overline{\tau}_{iso} \\ -\overline{\tau}_{iso} & \text{if } \overline{\tau}_{des} < -\overline{\tau}_{iso} \end{cases} \tag{10}$$

Eqns 9 and 10 represent the normalized feedback system, with all parameters and system behaviour expressed in dimensionless units. For example, the normalized proportional controller gain parameter is given by $\overline{K}_p = \dfrac{K_p}{\left(\frac{I}{t_{SM}^2}\right)}$ and the normalized response time is given by $\overline{t}_{resp} = \dfrac{t_{resp}}{t_{SM}}$. The *force capacity*, represented by $\overline{\tau}_{iso} = \dfrac{\tau_{iso}}{\left(\frac{I\theta_r}{t_{SM}^2}\right)}$, is the only free parameter in this model. Unlike in the scaled models, parameters related to animal size, time delay, and perturbation magnitude are all unity in this model. In all our models, including this one, controller gains are not free parameters as they are fixed at their optimal values (which we determined by optimization).

This model applies to both swing and posture tasks. We focus on studying the swing task in the main body of this manuscript and the posture task in the supplementary material (section S3 in S1 Text). We also present an additional analysis using Bode plots in the supplementary material that explains how time delays limit controller gains in a linear feedback system (section S2 in S1 Text).

We simulated and optimized the normalized swing and posture task models using the same protocols and methods as described for the scaled models, except for the following differences. For the swing task, we set the initial conditions for plant state $[\overline{\theta}_0, \dot{\overline{\theta}}_0]$ to [0, 0], and the target state $[\overline{\theta}_r, \dot{\overline{\theta}}_r]$ to [1, 0]. We first simulated the normalized model without a force capacity limit ($\overline{\tau}_{iso} = \infty$), and then for a range of force capacity limits ($\overline{\tau}_{iso} = 0.001 : 0.001 : 0.25$), and optimized the controller gains ($\overline{K}_p$ and $\overline{K}_d$) for each force capacity limit to achieve the fastest response time. We performed the derivations to normalize the equations by hand, and simulated the normalized feedback system using MATLAB's Simulink toolbox.

## 3. Results

### 3.1 Normalized feedback control system with time delays and force capacity limits

Optimizing without force capacity limits ($\overline{\tau}_{iso} = \infty$) and with a dimensionless time delay of unity, we found that controller gains of $\overline{K}_p=0.1617$ and $\overline{K}_d=0.6343$ produced the fastest response time of 7.09, all in normalized units. The magnitude of this response time means that it took ~7x the duration of the sensorimotor time delay to move the physical system from its original state to settle on its new target position. The torque during this simulation reached a maximum value of 0.1617, equal to the value of $\overline{K}_p$. This maximum simulated torque means that while there was no limit to available force, the control system was only able to use 0.1617 units of dimensionless torque at a maximum. The torque curve has a fast acceleration phase which reaches a maximum of 0.1617, followed by a slower braking phase which reaches a peak torque of -0.0950 (Fig D in S1 Text), due to the proportional and derivative terms dominating at different times in the response. The optimal gains and system performance would be identical with a force capacity of $\overline{\tau}_{iso} = 0.25$, for example, as it would be without any limit to force capacity. Thus, we next systematically decreased $\overline{\tau}_{iso}$ from 0.25 to 0, keeping time delay constant at unity, and optimized controller gains to achieve the fastest response times (Fig 2). The relationship between normalized response times ($\bar{t}_{resp}$) and force capacity limits ($\overline{\tau}_{iso}$) showed three distinct regions indicated by the grey vertical lines: a high region ($\overline{\tau}_{iso} > 0.1610$), a middle region ($0.079<\overline{\tau}_{iso}<0.1610$), and a low region ($0.009<\overline{\tau}_{iso}<0.079$) (Fig 2 top). In the high region, response times do not change with force capacity, as they are purely delay-limited; the peak torques do not reach the force capacity limits. In the middle region, response times begin to increase gradually with force capacity

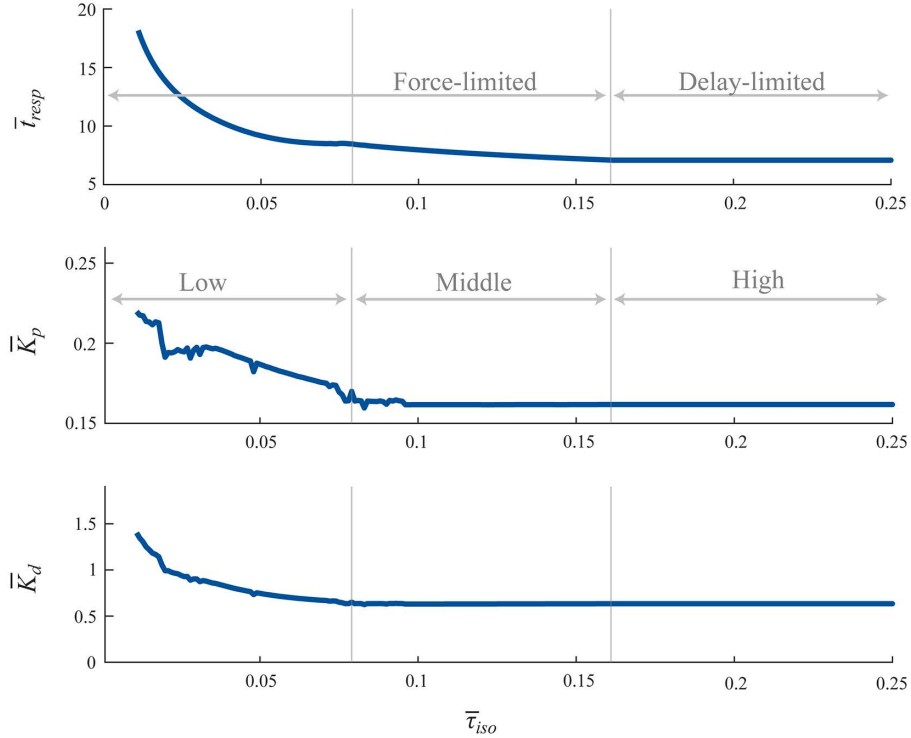

**Fig 2. Normalized model—swing task response times vs. force capacity limits.** The fastest normalized response times ($\bar{t}_{resp}$) and the optimal controller gains ($\overline{K}_p$ and $\overline{K}_d$) that produced these responses for a range of force capacity limits ($\overline{\tau}_{iso}$). The $\bar{t}_{resp}$ vs. $\overline{\tau}_{iso}$ relationship reveals three regions demarcated by the grey vertical lines. Response times are delay-limited in the high region, and force-limited in the middle and low regions. Note that there is some variability in the optimal controller gains in the middle and low regions due to multiple local minima that have values very close to the global minimum. This variability does not significantly affect the response time.

limits, as positive torques are clipped by the force limits, but negative torques are not. In the low region, both positive and negative torques are clipped by the force limits, causing response time to increase more rapidly. The supplementary material provide a more detailed explanation of the normalized feedback control system (section S3 in S1 Text).

This analysis shows that for feedback control systems with both time delays and actuator force capacity, there are two operating ranges: a delay-limited range and a force-limited range. Consider first the extremes. Without force capacity limits or delays, infinitely high gains can produce instant response times. However, if either the force capacity was set to 0, or if time delays were infinitely long, response times would be infinitely long. The delay-limited range consists of the high force capacity region, where only time delays affect response time. The force-limited range consists of the middle and low force capacity regions, where force capacity limits begin to also limit response time. Extrapolating to animals, this analysis indicates that for an animal to respond quickly, it requires both strong muscles and short sensorimotor delays. Deficiencies in either factor can slow the animal's ability to respond quickly. Whether an animal is delay-limited or force-limited would depend on the relative magnitudes of factors that affect the perturbation response—such as the moment of inertia of the body segments being moved, the sensorimotor delays, the size of the perturbation, and the muscle force capacity. These effects are seen in the normalization factor $\frac{I\theta_r}{t_{SM}^2}$ where increases in inertia or perturbation size will cause a shift towards the force-limited range even if the absolute force capacity is unchanged. And an increase in $t_{SM}$, which is in the denominator of this factor, will cause a shift away from the force-limited range and towards the delay-limited range, again even if muscles have no increase in absolute force capacity. We found similar results for the posture task (section S3 of S1 Text). We also compared the predictions from the normalized model to the results from the scaled model simulations and describe the results in section S4 of S1 Text.

## 3.2 Scaling of perturbation response times for scaled models

**3.2.1 Swing task.** Feedback control performed poorly when compared to feedforward control in the swing task (Figs 3 and 4). For a movement size of 30°, the fastest response times under feedback control scaled as $199M^{0.21}$ ms, compared to $62M^{0.24}$ ms under feedforward control (Fig 3c). This is because the feedback controllers were delay-limited, and unable to utilize a significant portion of their muscle force capacity due to the stability limitations imposed by long sensorimotor delays (Fig 4a). That the models were delay-limited across animal size means that the sensorimotor delay scaling determines the feedback control response time scaling—the feedback control scaling exponent is 0.21 because the sensorimotor delay exponent is 0.21. Because the exponents are the same, the feedback control response times are a constant multiple of the sensorimotor time delay and determined by the ratio of scaling coefficients (198.9/31). This equates to response time being 6.4x the duration of the time delay, which is close to the 7x prediction of the much simpler normalized model above. The feedforward control response is a combination of an initial deadtime during which the limb falls under gravity, and a bang-bang controlled repositioning movement under maximal muscle torques ($\tau_{iso}$). While the initial deadtimes scaled with sensorimotor delay ($M^{0.21}$), the bang-bang movement scaled approximately with inertial delay ($M^{0.28}$) [37]—together causing feedforward response times to scale with $M^{0.24}$. Therefore, the ratio of feedback response times to feedforward response times gets smaller with animal size, ranging from 4.1 in smaller animals to 2.3 in larger animals (Fig 4b). Table 1 lists the input parameters and the optimized output parameters in the swing task.

Larger animals also have slower characteristic movements, giving them longer to complete a perturbation response. After More & Donelan (2018) and Thangal & Donelan (2020), we estimated the shortest time available to complete a perturbation response for animals of different sizes as the swing duration at maximum sprint speed [24,37]. Table 1 provides the coefficients for this scaling relationship—we refer readers to the original papers for its derivation. Response times under feedback control exceeded swing duration at maximum sprint speed for all animal sizes, while feedforward control did not (Fig 4c).

**3.2.2 Posture task.** While the feedback controllers in the posture task were able to use more of the available torque than in the swing task, they still performed poorly compared to feedforward control (Figs 5 and 6). For a 0.21

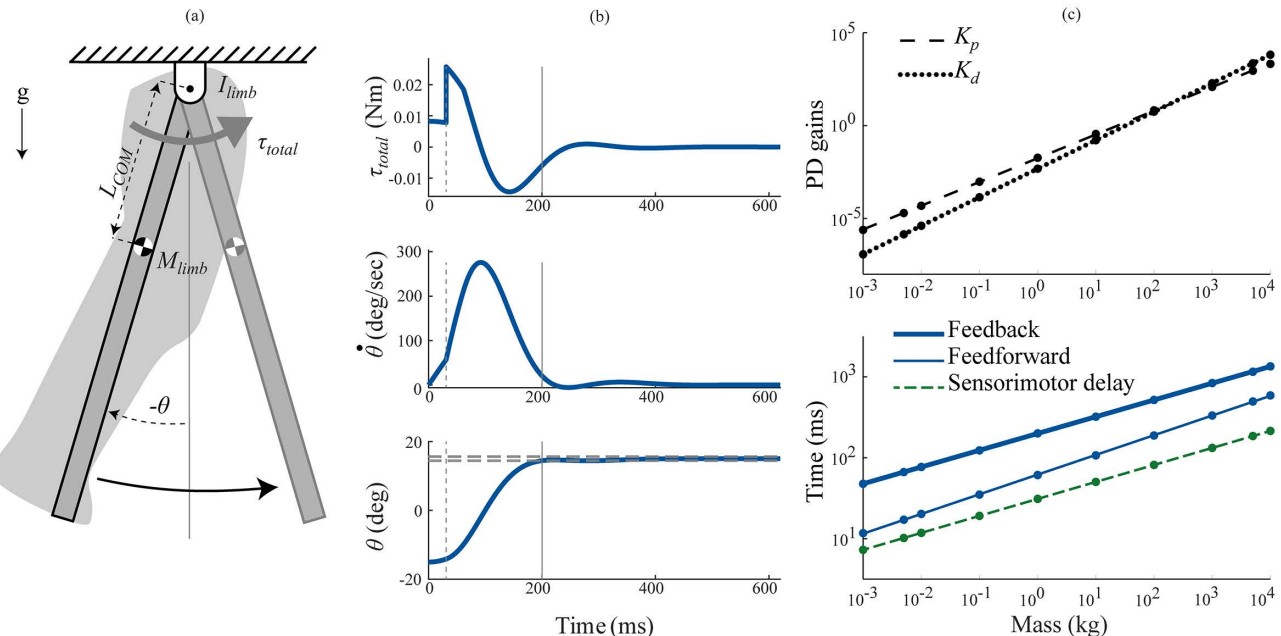

**Fig 3. Scaled model—swing task control.** The swing task simulates repositioning of the forelimb in response to an early swing phase trip. (a) We modeled the plant as a distributed mass pendulum actuated by torques generated by a PD controller. (b) Torque ($\tau_{total}$), angular velocity ($\dot{\theta}$) and angle ($\theta$) profiles in the swing task for a one kg animal and a 30° movement under feedback control. The grey dashed vertical line at 31 ms represents the initial sensorimotor delay period and the gray vertical line at 200 ms represents response time, computed as the settling time of the angle curve with a 2% threshold (grey dashed horizontal lines). (c) Log-log plots for the scaling of controller gains (top) and the response times (bottom) under feedback control (thick blue line), feedforward control (thin blue line) and sensorimotor delays (thin dashed green line). The points denote the actual values obtained through optimization, while the lines denote the power law fit.

dimensionless velocity perturbation, the fastest response times under feedback control scaled as $239M^{0.22}$ ms, compared to $95M^{0.28}$ ms under feedforward control (Fig 5c). Other than for the largest animals, feedback control in the posture task was delay-limited rather than force-limited (Fig 6a). The feedback control response time was approximately 7.7x the duration of the sensorimotor time delay across animal size. Under feedforward control, initial deadtime scaled with sensorimotor delay ($M^{0.21}$), and the bang-bang repositioning movement scaled with inertial delay ($M^{0.35}$) [37]—together causing feedforward response times to scale with $M^{0.28}$. The ratio of feedback response times to feedforward response times ranged from 3.8 in smaller animals to 1.3 in larger animals (Fig 6b). Table 2 lists the input parameters and the optimized output parameters in the posture task.

Feedback control response times in the posture task for the 0.21 dimensionless velocity perturbation also exceeded available time for all animal sizes (Fig 6c). We estimated the time available to complete the perturbation response by defining it as the time required to fall under gravity to the ground from a height equal to the leg length for each animal size, which scaled as $182M^{0.19}$ ms. Feedforward response times for the largest animals also exceeded available time (Fig 6c). Feedforward response times get longer with perturbation size, while feedback response times do not (unless muscle force capacity limits are exceeded). If we had used stronger perturbations, we would generally have seen longer feedforward response times while feedback response times remained constant, resulting in smaller feedback to feedforward response time ratios (Fig 6b), and lighter animals exceeding available movement time.

We have included some additional analyses in S1 Text. Section S5 in S1 Text provides a breakdown of the components of the overall torque applied by the feedback controller in the scaled model simulations. Section S6 of S1 Text compares the scaled model results to in-vivo perturbation responses for cats and humans reported in the literature.

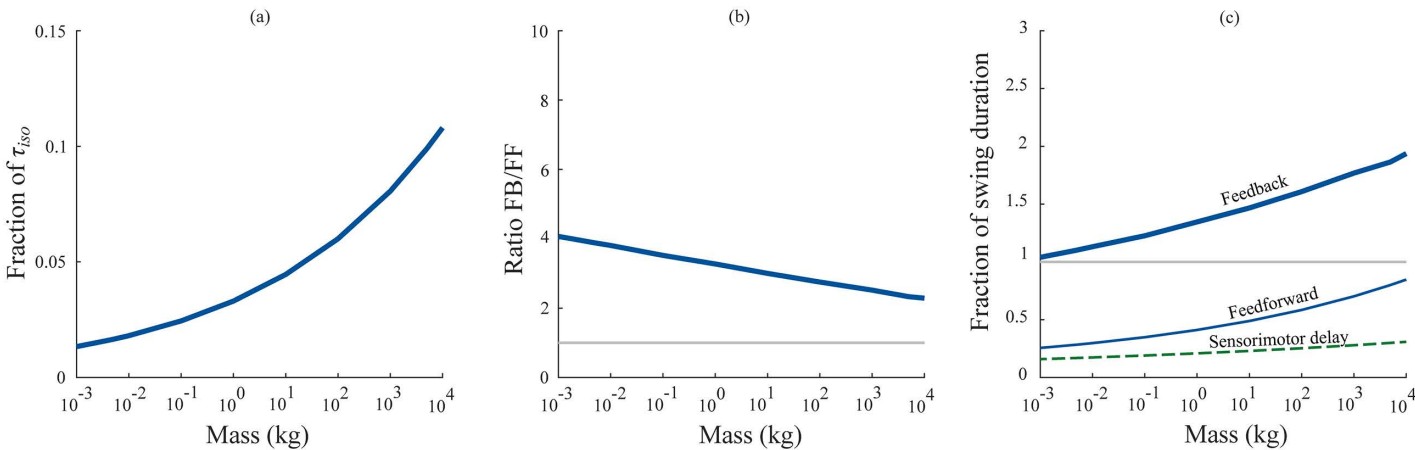

**Fig 4. Swing task—comparison of feedforward and feedback response times.** (a) The fraction of torque available based on muscle force capacity ($\tau_{iso}$) used by the feedback controller in the swing task across animal size. (b) The ratio of feedback control response times to feedforward control response times. (c) Fraction of swing duration at maximum sprint speed required to perform a corrective movement using feedback control (thick blue line), feedforward control (thin blue line) and sensorimotor delays (thin dashed green line).

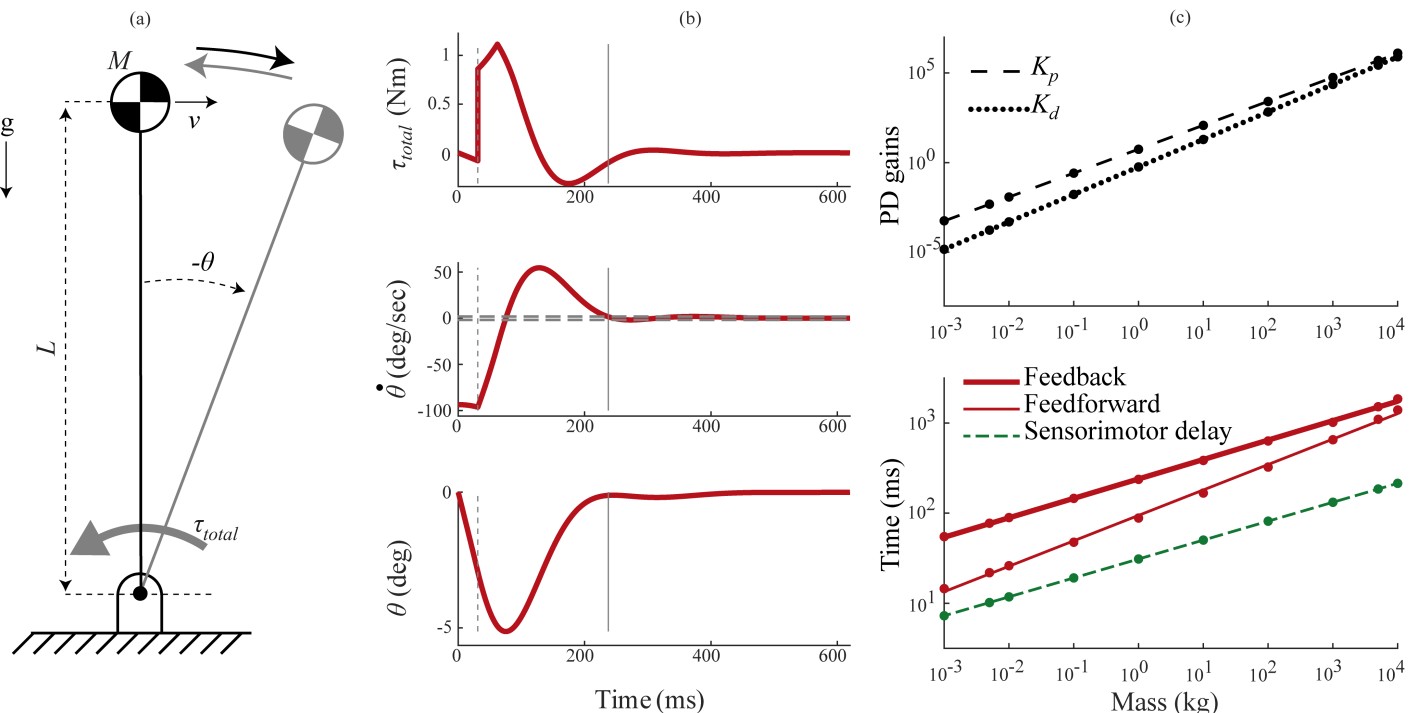

**Fig 5. Scaled model—posture task control.** The posture task simulates whole body posture control after a push forward in the sagittal plane. (a) We modeled the plant as a point mass pendulum actuated by torques generated by a PD controller. (b) Torque ($\tau_{total}$), angular velocity ($\dot{\theta}$), and angle ($\theta$) profiles in the posture task for a one kg animal, perturbed by a force causing an initial dimensionless velocity of 0.21, under feedback control. The grey vertical dashed line at 31 ms represents the initial sensorimotor delay period and the gray vertical line at 239 ms represents response time, computed as the settling time of the angular velocity curve with 2% thresholds (grey horizontal dashed lines). (c) Log-log plots for the scaling of the controller gains (top) and the response times (bottom) under feedback control (thick red line), feedforward control (thin red line) and sensorimotor delays (thin dashed green line). The points denote actual values obtained through optimization, while the lines denote the power law fit.

PLOS Computational Biology

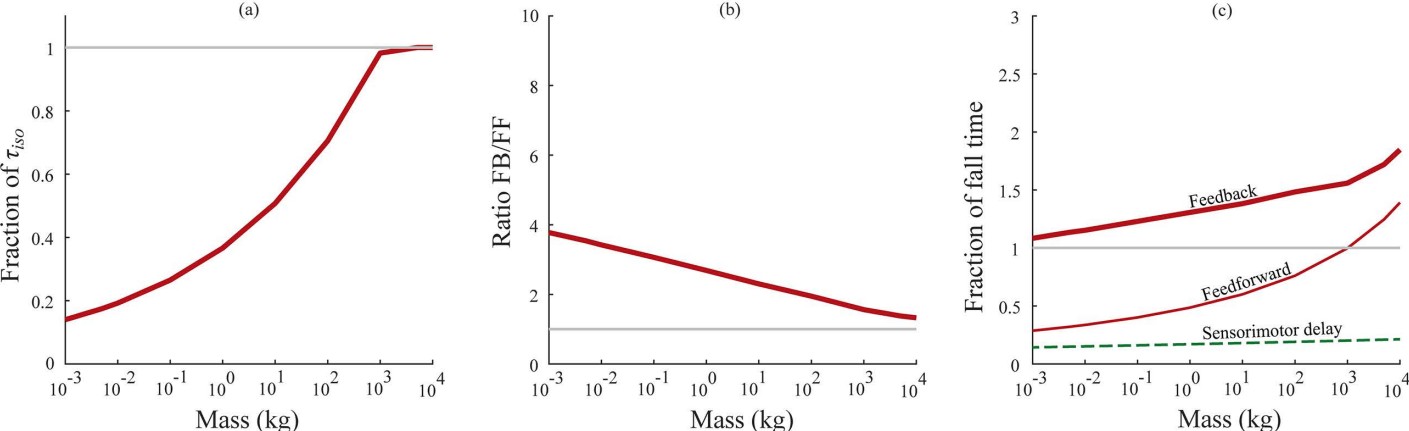

**Fig 6. Posture task—comparison of feedforward and feedback response times.** (a) The fraction of maximum isometric torque ($\tau_{iso}$) utilized by the feedback controller in the posture task across animal size. (b) The ratio of feedback control response times to feedforward control response times. (c) Fraction of available time (time taken to fall to the ground under gravity) required to perform a corrective movement using feedback control (thick red line), feedforward control (thin red line) and sensorimotor delays (thin dashed green line).

## 4. Discussion

In this study, we developed simple computational models to represent fast perturbation responses in animals, which are subject to sensorimotor delays and muscle force capacity limits. We quantified how response times scaled with animal size under feedforward and feedback control. Previous publications from our group had quantified the scaling of sensorimotor delays (signal transmission and processing time delays in the monosynaptic stretch reflex) [24], and inertial delays (movement time required to physically reposition body segments as part of the response to a perturbation) [37]. In this paper, we evaluated how the type of control used—feedforward vs. feedback control—further affected response times. We found that while feedforward control can fully activate muscles and produce fast responses, long sensorimotor delays required feedback control to use low gains to ensure stability, allowing only a fraction of the muscles' force capacity to be utilized. That is, the effectiveness of feedback control within the size range of terrestrial mammals is predominantly delay-limited rather than force-limited. Feedback response times were about seven times longer than the duration of sensorimotor delays, and at least one and a half times longer than feedforward response times across animal size, for both the swing task and posture task. Feedback response times exceeded available movement time for all animal sizes, while feedforward response times did so only for the largest animals in the posture task (Figs 4c and 6c). Feedback control does not seem effective for perturbations that require very fast responses in animals of any size. Animals are able to locomote effectively and perform fast perturbation responses in real world settings, indicating that their neural controllers are leveraging feedforward control in these contexts, or able to compensate for the limitations posed by sensorimotor time delay under feedback control.

To reach these conclusions, we had to make several assumptions and simplifications. We have carried over assumptions made in the estimation of parameters in previous publications, such as for the scaling of sensorimotor delays [24] and inertial delays [37]. For example, for the scaling of sensorimotor delay, More and Donelan (2018) had assumed that sensing delay, synaptic delay, and neuromuscular junction delay are constant across animal size based on limited information in the literature. In calculating muscle force capacity limits, we assumed that the triceps (swing task) and the ankle extensors (posture task) are the dominant muscles involved in moving their respective joints, that their antagonistic muscles scale similarly, and that the isometric stress produced by mammalian muscle is constant at 20 N/cm² [50]. We have approximated gradual changes in muscle force with fixed electromechanical and force generation delays, resulting

in instantaneous changes in our modeled muscle torques that are non-physiological. While this is not a limitation of some dynamic models of muscle behavior (e.g., Hill type), we chose not to incorporate them for two reasons: because it keeps our models simple and thus tractable, and because there is a lack of data to accurately scale muscle-tendon architecture across the size range of terrestrial mammals. Although animal limbs are multi-jointed and multi-muscled, we used pendular models to represent body segments, and a single pair of opposing muscles to actuate them [36,47]. We have also not considered several animal features that have been shown to change with animal size such as posture, limb stiffness, joint damping, and sensor accuracy; evaluating the effects of these features would require more complete neuromusculoskeletal models [59–61]. We used a PD feedback controller as it represents full state feedback in mechanical systems. It is simple to implement, simple to interpret, and has been successfully used in the literature to model control of locomotion [23,62–65]. The actual control strategies implemented within the spinal reflex pathways are undoubtedly more complex, involving several control pathways with varying time delays, and remains a topic of research [19,66–68]. While we used bang-bang actuation for our minimum-time feedforward controller to find the fastest response benchmark, feedback controllers can also achieve bang-bang actuation. Feedback control methods such as Model Predictive Control and Smith predictors can compensate for time delays and generate bang-bang actuation [69,70]. Here, we assumed that the spinal-level synapses that represent the feedback controller do not encode these complex algorithms and use the simpler and more straightforward proportional-derivative control instead.

While we have modeled our perturbation responses as goal-directed repositioning movements, perturbation responses during rhythmic locomotion also involve spinal neural oscillators and passive mechanisms. Preflexes are passive viscoelastic responses to perturbations that act instantaneously and without neural input, although they are modulated by the active state of muscles and joint positions during movement [71]. Running guinea fowl encountering obstacles maintain rhythmic feedforward control of proximal leg joints, while distal joints respond preflexively [68,72,73]. Humans adjust limb stiffness in anticipation of floor stiffness changes during hopping and running, relying on feedforward tuning and preflexive responses [74,75]. And cockroaches subjected to impulsive lateral perturbations generate peak ground reaction forces far faster than can be explained by neural reflexes, highlighting how intrinsic mechanical properties and pre-tuned musculoskeletal dynamics can mediate instantaneous responses [76,77]. Kuo (2002) described how central pattern generators and reflex circuits work together to form a state estimator which can optimally compensate for both disturbances and sensorimotor noise [7]. We have not modeled these CPG and preflexive control mechanisms, and considered only upright terrestrial quadrupedal mammals, to simplify modeling choices.

Given these assumptions, as well as the nature of any purely modeling approach, these results are estimates that need to be tested experimentally. We have tried to estimate the limits of neural control by considering sensorimotor delays from the longest reflex pathways (a distance equal to twice the animal's leg length), which used the least neural computation (a single synapse), and compared response times to the shortest available movement times (e.g., swing duration at maximum sprint speed). Using these highly simplified models to evaluate the limits of performance as a function of animal size, we have made predictions about the complex processes of animal locomotor control. One candidate experimental approach to test our predictions would be in-vivo experiments where the delays in sensory feedback or transmission of motor commands can be systematically manipulated while studying the response of the nervous system [78]. Several studies have investigated the effects of removing sensory feedback on locomotor behavior in animal models [2,9,63,79]. Instead of removing sensory feedback, one might decrease the nerve conduction velocity using cooling techniques to increase the sensorimotor delays in the reflex [80,81]. Such experiments would have to devise ways to cool only the nerves while minimizing changes to the muscle, perhaps by using nerve cuffs [82]. The results from various species could then be compiled to evaluate various neuromechanical control architectures that integrate feedforward and feedback control in different ways [83,84].

Several studies have investigated the compensatory mechanisms that animals could be using to overcome the drawbacks caused by sensorimotor delays. The models in this paper use the simplest method to deal with time delays in a

feedback system, which is to reduce the controller gains—this slows down the system's performance but prevents instability. Weiland et al. (1986) artificially introduced time delays into the reflex loops that control the femur-tibia joint in stick insects, and showed that increasing time delays caused instability in the form of tremors [78]. Tuthill et al. (2018, 2021) proposed that the supraspinal motor centers act to inhibit activity—reduce the controller gains—in reflex circuits in order to keep them stable despite the sensorimotor delays, because severing supraspinal inputs results in tremors in animals [85,86]. Blondiaux et al. (2024) show that inaccurate compensation for long latency feedback delays can explain essential tremor [87]. A more computationally intensive method to compensate for delays is to use neural prediction—the nervous system could implement internal models that can estimate the present state of the body from time delayed sensory feedback. Miall et al. (1993) suggested that one of the functions of the cerebellum is to act as a Smith predictor, an internal model specifically designed to compensate for time delays in feedback control systems [70]. Rasman et al. (2021, 2023) used a robotic balance simulator to artificially introduce time delays into the sensory feedback signals involved in the neural control of upright standing balance [88,89]. They showed that with training, the internal models involved in upright standing can learn to compensate for even 400 ms of additional time delay, while the human body inherently has only 100–160 ms of sensorimotor delays. The nervous system likely encodes several compensatory strategies for sensorimotor delays with varying levels of complexity, with simple and fast strategies mediated through shorter reflex pathways, and slower and more complex strategies mediated through higher (brain stem, cortical, cerebellar) pathways [90].

While we found that sensorimotor delays are detrimental to the control of fast perturbation responses, other studies have pointed out that the nervous system could use time delays to its benefit in certain control situations. Studies have shown that time delayed positive feedback can help stabilize reflexive control pathways, and aid in rhythmic locomotion [19,62,64]. Nishikawa et al. (2007) show that time delays increase gains at the resonant frequency of a control system, which could assist in the control of rhythmic movements [68]. Besides time delays, the nervous system also must deal with other drawbacks such as noise, muscle force capacity limits, and limited sensory resolution. Milton and colleagues suggest that the nervous system is able to use the interaction between these various drawbacks to simplify neural control [35,91–94]. However, during our simulations, delays did not provide any benefits. For example, if an animal trips during early swing phase while running at high speed, it would be best to react immediately, fully activating its muscles to rapidly and accurately place its foot forward and regain stability. Irrespective of the control strategy, the animal would lose the time required to sense the perturbation, compute a response strategy and transmit action potentials through the reflex pathways. Under feedback control, the animal is further limited, as it must use low gains in order to ensure a stable response, while the corrective movement might not be quick enough to prevent a fall. Under this scenario, and others like it, delays appear to only negatively affect the control of movement.

One implication of our findings is that the compensatory mechanisms used by animals to overcome the disadvantages caused by long sensorimotor delays may vary with size. Previous studies in small animals such as guinea fowl [72,95] and cockroaches [60,76,77] have shown that they rely more on feedforward control than feedback control for fast locomotion, and utilize the inherent mechanical properties of their musculoskeletal system (preflexes) to compensate for perturbations [71]. This emphasis on feedforward control over feedback makes sense for lessening the consequences of time delays that we have shown here, but feedforward control has its own challenges because accuracy and speed are susceptible to errors in modeling one's own body, and the perturbations that are being applied. Garcia et al. (2000) showed that the limbs of small animals are overdamped while the limbs of larger animals are underdamped [60]. Overdamping will help prevent overshoot or oscillations when repositioning the body making small animals less sensitive to modeling errors in feedforward control. This is a candidate explanation for why smaller animals emphasize feedforward control, which can produce faster responses than feedback control while overdamping still ensures stable movement. Larger animals would not benefit from this damping—their underdamped bodies would be less tolerant to imperfect feedforward control with even small modeling errors resulting in overshooting and oscillations. Therefore, larger animals may emphasize feedback control to ensure accuracy and stability. Our results here further support different control strategies for small and large

animals by showing that uncompensated feedback control is up to four times slower than feedforward control in the smallest animals, but only about two to one and a half times slower in the largest animals.

If larger animals could compensate for sensorimotor delays and reduce feedback response times to be shorter than available movement time, feedback control could become viable for fast perturbation responses. Larger animals do not have highly damped joints, but synaptic delays make up a smaller proportion of overall response time [24]. These relatively shorter synaptic delays might allow larger animals to rely on more computationally intensive control strategies that combine feedback control with internal models that compensate for delays [70]. Our results show that as animal size increases, delayed feedback control can use more of the available muscle force capacity, narrowing the performance advantage that feedforward control has over feedback control. Effective feedback control would ensure that the underdamped limbs in larger animals are repositioned accurately without oscillations and overshoot/undershoot errors, and when combined with the computational options available when synaptic delays are relatively short, may be a more viable control strategy for larger animals.

## Supporting information

**S1 Text. Supplementary material.** In the document, we have provided detailed derivations for the equations in the paper, and described secondary analyses that we performed to evaluate our results. The document has the following sections:. S1. Delays in the feedforward and feedback pathways. S2. Bode plot analysis of a linear feedback control system. S3. Normalized feedback control system with time delays and actuator force capacity—detailed derivations and analyses. S4. Normalized feedback model predictions vs. scaled model simulation results. S5. Components of total applied torque under feedback control. S6. Comparing swing and posture task responses to in-vivo perturbation studies.
(DOCX)

## Author contributions

**Conceptualization:** Sayed Naseel Mohamed Thangal, J. Maxwell Donelan.

**Data curation:** Sayed Naseel Mohamed Thangal.

**Formal analysis:** Sayed Naseel Mohamed Thangal, C. David Remy, J. Maxwell Donelan.

**Funding acquisition:** Sayed Naseel Mohamed Thangal, J. Maxwell Donelan.

**Investigation:** Sayed Naseel Mohamed Thangal, Heather L. More, C. David Remy, J. Maxwell Donelan.

**Methodology:** Sayed Naseel Mohamed Thangal, Heather L. More, C. David Remy, J. Maxwell Donelan.

**Project administration:** Sayed Naseel Mohamed Thangal, J. Maxwell Donelan.

**Resources:** Sayed Naseel Mohamed Thangal, J. Maxwell Donelan.

**Software:** Sayed Naseel Mohamed Thangal, C. David Remy, J. Maxwell Donelan.

**Supervision:** C. David Remy, J. Maxwell Donelan.

**Validation:** Sayed Naseel Mohamed Thangal, J. Maxwell Donelan.

**Visualization:** Sayed Naseel Mohamed Thangal, J. Maxwell Donelan.

**Writing – original draft:** Sayed Naseel Mohamed Thangal, J. Maxwell Donelan.

**Writing – review & editing:** Sayed Naseel Mohamed Thangal, Heather L. More, C. David Remy, J. Maxwell Donelan.

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
