## [Decision Letter · Decision Letter 0]

9 Mar 2025

PCOMPBIOL-D-24-01586

Effects of sensorimotor delays and muscle force capacity limits on  the performance of feedforward and feedback control in animals of different sizes

PLOS Computational Biology

Dear Dr. Mohamed Thangal,

Thank you for submitting your manuscript to PLOS Computational Biology. Your paper was reviewed by the editorial team and by two expert reviewers. The reviewers appreciated that the paper contributes some interesting insights about how motor control strategies differ across scales. However, the reviewers did raise several concerns. While some of these issues are clearly addressable (e.g. improving/clarifying presentation), there were other substantial concerns that may not be. In particular, Reviewer 2 raised some important and fundamental issues with the analysis and conceptual framing of the parts of the paper that focus on feedforward control. Both reviewers also expressed concerns about whether the models might be overly simplistic. It's not entirely clear whether these issues can be resolved within the scope of changes that would constitute a "major revision". However, given that there is clear merit in parts of the paper, if you believe that all of the concerns can be adequately addressed, then we would be willing to consider a revised version of the manuscript along with responses to the reviewer's comments.

Please submit your revised manuscript within 60 days May 09 2025 11:59PM. If you will need more time than this to complete your revisions, please reply to this message or contact the journal office at ploscompbiol@plos.org. Please include the following items when submitting your revised manuscript:

We look forward to receiving your revised manuscript.

Kind regards,

Adrian M Haith

Academic Editor

PLOS Computational Biology

Lyle Graham

Section Editor

PLOS Computational Biology

**Journal Requirements:**

At this stage, the following Authors/Authors require contributions: Sayed Naseel Mohamed Thangal, Heather L More, and Max Donelan. Please ensure that the full contributions of each author are acknowledged in the "Add/Edit/Remove Authors" section of our submission form.

**Reviewers' comments:**

Reviewer's Responses to Questions

Reviewer #1: Sensorimotor delays and muscle force capacity scaling for control of movement PloSCompBiol

The authors provide two simple models for testing the relative efficacy of feedback versus feedforward control of (1) a (fore)limb swing perturbation recovery task versus (2) a body CM support (hind)limb recovery task to evaluate how body size (mass) affects these two differing control strategies. In doing so, they build on earlier experimental and modeling work that evaluated how sensorimotor delays and inertial effects scale with animal size. Their modeling here shows that small animals may well benefit from feedforward control of perturbations for recovery from fast movements; whereas, both small and larger animals face challenges due to sensorimotor delays in responding to fast perturbation challenges under feedback control. The introduction does a nice job of providing a broader context for their modeling study. However, while I generally enjoyed reading their paper, I was disappointed by the sloppy organization of the methods and results, with figures out of sequence and several terms and parameters that were not defined or explained clearly. This made for a confusing read at times. I am not a controls engineer, so several detailed aspects of control theory and elements of their model are not ones that I can evaluate critically. But there are other areas (from the biological/biomechanics side) in which I am familiar, and certain specific aspects of their models need to be better justified and explained. With improved organization and clear explanation of terms, and definition of parameters, this will make a valuable paper to build on the lead (corresponding) author’s work. In general, the paper could benefit from being more concise (this also applies to the discussion, which is rather long and overly speculative in places).

My specific comments follow:

l. 56-57 – need citations to Daley & colleagues’ relevant papers here, presumably [7-11], or identify them more specifically through l. 64. Leaving the [7-11] citation at l. 65 makes it unclear which of the preceding statements related to the citations [7-11].

l. 63 & l. 68: it might be helpful to define what you mean by ‘preflexively’ or ‘preflexes’ for a general audience.

l. 106: I appreciate the citation to [29], but this (though the first) was a limited study on the scaling of stride frequency vs body size (based on relatively few species), which did not focus on the time duration for correcting a disturbance (stride or step period) – which is the point the authors wish to make here. I think making the link between stride frequency, step/stride duration, and response time to disturbances for control of movement is needed vs this brief statement and citation to [29]. Perhaps one or more other papers should be cited? Also, Heglund & Taylor (JEB, 1988) provides stronger evidence for the scaling of stride frequency than the Science [29] paper.

Fig. 1 (& Methods) – use of a bang-bang controller for feedward activation/control should be explained and refs cited.

I am most concerned by the non-physiological square-wave input torque control of this approach (1B). Authors should explain and justify this, beyond presenting the conditions at l. 197. (1A) The rapid rise in input torque for feedback control also seems non-physiological. Can the authors explain/justify this?

l. 182: “ des” is not defined – an equation at l. 187 is given but no description of what the parameter represents [it is defined on p. 17 at l. 294!]

p. 13: “ SM” is also not defined [it is eventually defined on p. 17 at l. 300!]

l. 235-236: Why are data for the scaling of the triceps muscle from Alexander et al 1988 used to parameterize shoulder muscles? Triceps acts mainly at the elbow in mammals, including humans. It is not a muscle that will swing the forelimb forward at the shoulder or support body weight related forces during stance (posture task). Deltoid, pectoralis supraspinatus would be more anatomically and functionally relevant.

L, 248-249: I was initially confused by the authors stating that the posture perturbation control task is achieved by the plantarflexors. This is because no clear explanation of the two tasks is provided at the outset of the paper. In defining and modeling these two tasks (swing vs posture) the authors need to make clear that the swing task concerns a perturbation to the swing of the forelimb – controlled at the shoulder versus a posture task that involves control of a perturbation causing an angle change in the supporting hindlimb to control CM motion; making it clear that two different sets of muscles (and limbs) are involved in these two tasks.

This is clearly stated at l. 499-500 but needs to be made clear in Methods when first defining these tasks. But again, re: l. 499-500, the triceps do not control swing motion at the shoulder. I know of no evidence for this in any mammal.

Fig. 2 (p. 26!) is not referenced or discussed prior to Fig. 3 at l. 170. Much of the explanation of Fig. 2 in the legend is not clearly explained or justified as a result.

Fig. 3(B) p. 28: please explain justification for the instantaneous ramp in torque at 31 ms

l. 396: Also, why is Fig. 3 titled this way: “Fig 3. Scaled model—swing task responses under feedback control”? Feedforward control is also shown in comparison to feedback control.

l. 443: This also applies to Fig. 5 “Fig 5. Scaled model—posture task responses under feedback control”, which also shows a comparison of feedback vs feedforward control in panel C. Hence, figure titles for figs. 3 & 5 are inconsistent with what is shown.

l. 410: How was swing duration at maximum sprint speed determined? [19,27] are cited but referencing these studies does not explain how EITHER maximum sprint speed vs animal body mass or swing duration were calculated. [19,27] report sensorimotor and inertial delays – not maximum spring speeds or swing durations. Maximum speeds of terrestrial animals are notoriously noisy and difficult or unreliable to determine.

l. 435-437 “The ratio of feedback response times to feedforward response times ranged from four in smaller animals to one and a half in larger animals (Fig 6b).” This statement doesn’t match well what Fig. 6b shows, which by my eye ranges from <4 (~3.8?) to ~1.3 not 1.5.

l. 454-455: This statement is presumably in reference to Fig. 6C, but this is not made clear. Explicit reference to the figures would help make the presentation of your results clearer.

l. 505: “we used … a single pair of opposing muscles to actuate them [26,44].” I don’t follow this statement. No antagonistic pair of muscles is presented for controlling either the shoulder in the swing task or the ankle (plantarflexors) of the posture task. Only one set for each task.

l. 514-515: provide (textbook) references for these complex feedback controllers?

Paragraph from l. 519-534 is overly speculative, as the experimental techniques suggested are close to being unrealistic: “Experiments of this nature would be challenging to carry out in animals spanning the full size range of terrestrial mammals.” An understatement if there ever was one!

l. 564: I think I know what is meant by ‘sensory dead zones’, but some definition or clearer statement of what is meant would be helpful.

l. 584: define ‘damping ratio’

l. 600: “on more computational[ly intensive] control strategies”

l. 752-755: ref 56 is quite odd, with information irrelevant to the paper cited.

Reviewer #2: In this paper, Thangal et al carry out a computational modeling study investigating the role that sensory delay and actuator force saturation play in limiting the response times of a feedback controller. This study is carried out on two simulated plant models: a swinging limb in gravity and a postural balance task. They compare feedback (FB) control performance to a feedforward (FF) controller and conclude. Care is taken to dimensionally scale the system and control across an extremely wide size range.

First, let me focus on the feedback control conclusions. I think there is value in their feedback control analysis particularly for the scaled models. The main idea in the paper is simple: feedback is generally delay limited, and not actuator limited, in producing the “fastest” response to a perturbation. This is perhaps obvious at some scales, but it is not obvious how this will scale across a wide range of animal sizes nor was it obvious to me if or when one might see a crossover between force- vs. delay limitations. After reading the paper, this made sense but it was interesting and thoughtful. I will say that I don’t “believe” the exact cross over point because I think that one thing the model doesn’t take into account is that damping becomes dominant at lower amplitudes, for example, and smaller animals are often somewhat more sprawled. But the general trends and scaling relationships are nevertheless interesting and one can’t expect for a simplified model that scales from mice to elephants to get it all right for everything. That’s not the intent of the modeling. In short, this part seems a solid result and is the main strength of the manuscript, building nicely on lab’s prior work on scaling of feedback delay.

However, the FF control feels like a straw man. Normally, one thinks of a FF motor program as a set of motor commands that are driven e.g. by a CPG and not subject to sensory feedback. Of course most researchers including the authors recognize that this is an over simplification as CPGs are generally modulated in relation to ongoing sensory feedback, but the idea is that the time scales of that feedback are much longer than the strong initial responses that the system can provide to perturbations. This is beautifully illustrated in the paper they cite by Jindrich and Full showing that peak lateral leg forces due to an impulsive perturbation to a cockroach occur far faster than could be expected from neural reflexes, or the work by Daley’s group on perturbations to running bipedal birds, as also cited. However, these perturbation responses are caused by musculoskeletal interactions with the environment, e.g. passive viscoelastic responses of muscles—critically those responses can be tuned based on ongoing motor programs and therefore create a “preflexive” response. So,I had expected when reading the abstract, e.g. to see two muscles fighting each other, and increasing tonic antagonistic activation and showing using a muscle model how the muscular responses gets closer to maximum with a perturbation the higher the tonic antagonistic activation. Instead, the authors proposed an *EXACT* bang-bang controller that moves you from a perturbed position to the home position. FF control can’t do this. It requires feedback to exactly cancel out a perturbation in the way proposed. This is especially true for the unstable model. In other words, creating a pre-planned motor program that must be exactly tuned to the perturbation is not "FF control". Also, it is well known that the best way to do this in controls to achieve a minimum time response is typically bang bang at the actuator limits so it is a foregone conclusion that it will always hit the maximum force.

So, in sum, the FB control modeling is interesting and informative, but the lack of muscle dynamics and other passive mechanisms in the model (which are partially described in the intro) that integrate with a more appropriate form of FF control is a major weakness of the paper. The FF “control” described herein is more like a pre-programmed ballistic movement and so far as I can tell has nothing to do with rejecting perturbations. Ideally the FB controller would also include a muscle model.

I see a few ways of going — cutting FF altogether and having a much short communication, or dramatically improving the FF model. My idea above may be terrible so I’m not advocating a particular solution, but rather the model should capture the essence of preflexive perturbation responses.

Also, the FF/FB hypothesis has a rich history, much of which the authors cited. A couple other papers of interest that nicely describe FF/FB axis:

Revzen, S., Koditschek, D.E., Full, R.J. (2009). Towards Testable Neuromechanical Control Architectures for Running. In: Sternad, D. (eds) Progress in Motor Control. Advances in Experimental Medicine and Biology, vol 629. Springer, Boston, MA. https://doi.org/10.1007/978-0-387-77064-2_3

Holmes, Philip, et al. "The dynamics of legged locomotion: Models, analyses, and challenges." SIAM review 48.2 (2006): 207-304.

Minor:

the %OS graph being constant seems unnecessary.

There are a number of editing issues, but one that was particularly confusing was on p29 “We estimated time available to complete the pertubation response as the time…” do you mean “as a fraction of the time?

**Have the authors made all data and (if applicable) computational code underlying the findings in their manuscript fully available?**

The PLOS Data policy requires authors to make all data and code underlying the findings described in their manuscript fully available without restriction, with rare exception (please refer to the Data Availability Statement in the manuscript PDF file). The data and code should be provided as part of the manuscript or its supporting information, or deposited to a public repository. For example, in addition to summary statistics, the data points behind means, medians and variance measures should be available. If there are restrictions on publicly sharing data or code —e.g. participant privacy or use of data from a third party—those must be specified.requires authors to make all data and code underlying the findings described in their manuscript fully available without restriction, with rare exception (please refer to the Data Availability Statement in the manuscript PDF file). The data and code should be provided as part of the manuscript or its supporting information, or deposited to a public repository. For example, in addition to summary statistics, the data points behind means, medians and variance measures should be available. If there are restrictions on publicly sharing data or code —e.g. participant privacy or use of data from a third party—those must be specified.

Reviewer #1: Yes

Reviewer #2: **No:**The manuscript says the code is available in the supporting files, but I couldn't figure out how to download it (honestly didn't spend a ton of time trying). So if the paper is invited for revision it should be made clear that the code needs to be downloadable (and again, perhaps it was, but I didn't try to find it).The manuscript says the code is available in the supporting files, but I couldn't figure out how to download it (honestly didn't spend a ton of time trying). So if the paper is invited for revision it should be made clear that the code needs to be downloadable (and again, perhaps it was, but I didn't try to find it).

PLOS authors have the option to publish the peer review history of their article (what does this mean?). If published, this will include your full peer review and any attached files.). If published, this will include your full peer review and any attached files.

.

Reviewer #1: **Yes:**Andrew A. BiewenerAndrew A. Biewener

Reviewer #2: No

**Figure resubmission:**
---

## [Decision Letter · Decision Letter 1]

27 Aug 2025

PCOMPBIOL-D-24-01586R1

Effects of sensorimotor delays and muscle force capacity limits on the performance of feedback control in animals of different sizes

PLOS Computational Biology

Dear Dr. Mohamed Thangal,

Thank you for submitting your manuscript to PLOS Computational Biology. After careful consideration, we feel that it has merit but does not fully meet PLOS Computational Biology's publication criteria as it currently stands. Therefore, we invite you to submit a revised version of the manuscript that addresses the points raised during the review process.

Please submit your revised manuscript within 60 days Oct 27 2025 11:59PM. If you will need more time than this to complete your revisions, please reply to this message or contact the journal office at ploscompbiol@plos.org. Please include the following items when submitting your revised manuscript:

We look forward to receiving your revised manuscript.

Kind regards,

Lyle J. Graham

Section Editor

PLOS Computational Biology

**Reviewers' comments:**

Reviewer's Responses to Questions

**Comments to the Authors:**

**Please note that one review is uploaded as an attachment.**

Reviewer #1: Review PCOMPBIOL-D-24-01586R1

I appreciate the authors’ thorough and thoughtful responses to the questions and issues that I raised in my initial review, as well as the points raised by the other reviewer. Overall, the authors have done a very good job to improve the clarity and organization of their paper. The paper now reads much more clearly and is easier to follow the authors’ reasoning and modeling approach. I very much appreciate making clear the rationale for explaining why the presentation of the scaled models precedes the normalized feedback model in the methods but is reversed in the results. I also appreciate that more complex Hill-muscle models are not incorporated in the modeling approach given the lack of data across animal size. In summary, showing that differing control strategies for both tasks (swing and postural likely exist for small versus large animals, with the latter favoring feedback control whereas small animals that are overdamped can benefit from feedforward control for fast movements is a nice and valuable outcome of the authors’ modeling analysis.

I have only a few additional points to make/suggest:

l. 137: wrt the statement here “rendering their muscles proportionally weaker [26,38–41].” I would argue that Biewener, Science 1989 and/or Biewener, Science 1990, following from Alexander et al. [38], clearly noted the scale effects of size on muscle force capacity (matched to muscle x-area to maintain similar stresses across size), and should be included in this reference list. Also, I do not believe Heglund et al. 1979 Science [40] explicitly address the matter of relatively weaker muscles, in the context of examining SFreq scaling and A.V. Hill’s analysis of muscle power scaling. So, unclear why it is in the list.

L. 161: suggest adding “a forelimb swing leg repositioning task..” and l. 162 “a hindlimb posture recovery task under the control of ankle…” to make it quite clear that two different limbs (and joints) are being modeled for the swing and postural control tasks.

l. 283-288: I appreciate that Alexander et al. [38] only report data for the triceps muscle scaling – and yes, the long head of triceps does cross the shoulder to assist in shoulder flexion or retraction. HOWEVER, the moment arm reported by [38] for triceps is at the elbow joint, not the shoulder. This adds to the problematic approach of using triceps muscle scaling data for analysis of the shoulder swing task. This needs to be addressed and further clarified in revision; otherwise, readers would assume the moment arm scaling is for the shoulder, but not what [38] reports. In this context, I think you might consider a sensitivity analysis for the moment arm scaling relationship of what would be appropriate for shoulder muscles. Note that Biewener Science 1990 reports the scaling of antigravity muscle moment arms across forelimb joints for mammals of different size to be 0.44, slightly greater than 0.41 for the triceps at the elbow reported by Alexander et al [38]. In any event, you might consider how sensitive your results are to differing moment arm scaling relationships – perhaps 0.38 to 0.44, spanning 0.41 for your shoulder swing task model.

Reviewer #2: Review uploaded as attachment.

**Have the authors made all data and (if applicable) computational code underlying the findings in their manuscript fully available?**

The PLOS Data policy requires authors to make all data and code underlying the findings described in their manuscript fully available without restriction, with rare exception (please refer to the Data Availability Statement in the manuscript PDF file). The data and code should be provided as part of the manuscript or its supporting information, or deposited to a public repository. For example, in addition to summary statistics, the data points behind means, medians and variance measures should be available. If there are restrictions on publicly sharing data or code —e.g. participant privacy or use of data from a third party—those must be specified.requires authors to make all data and code underlying the findings described in their manuscript fully available without restriction, with rare exception (please refer to the Data Availability Statement in the manuscript PDF file). The data and code should be provided as part of the manuscript or its supporting information, or deposited to a public repository. For example, in addition to summary statistics, the data points behind means, medians and variance measures should be available. If there are restrictions on publicly sharing data or code —e.g. participant privacy or use of data from a third party—those must be specified.

Reviewer #1: Yes

Reviewer #2: Yes

PLOS authors have the option to publish the peer review history of their article (what does this mean?). If published, this will include your full peer review and any attached files.). If published, this will include your full peer review and any attached files.

.

Reviewer #1: **Yes:**Andrew A. BiewenerAndrew A. Biewener

Reviewer #2: **Yes:**Noah CowanNoah Cowan

**Figure resubmission:**
---

## [Decision Letter · Decision Letter 2]

5 Jan 2026

PCOMPBIOL-D-24-01586R2

Effects of sensorimotor delays and muscle force capacity limits on the performance of feedforward and feedback control in animals of different sizes

PLOS Computational Biology

Dear Dr. Mohamed Thangal,

Thank you for submitting your manuscript to PLOS Computational Biology. After careful consideration, we feel that it has merit but does not fully meet PLOS Computational Biology's publication criteria as it currently stands. Therefore, we invite you to submit a revised version of the manuscript that addresses the points raised by Reviewer 2.

We look forward to receiving your revised manuscript.

Kind regards,

Lyle J. Graham

Section Editor

PLOS Computational Biology

**Journal Requirements:**

**Reviewers' comments:**

Reviewer's Responses to Questions

**Comments to the Authors:**

Reviewer #1: The authors have addressed my minor concerns, compared with those raised by Reviewer #2. I will leave it to Reviewer 2, who's expertise in controls engineering related to neural sensory-motor control is considerably greater than mine. The authors have appeared to respond in depth to the several critical points of Rev #2.

Reviewer #2: I think the authors have addressed the major concerns raised in the previous review. The intro is now clear and comprehensive. The manuscript is clear and detailed, especially with the Supplementary Material. Our remaining concerns are minor.

1. In Figure 1, I don’t understand why the delay is placed in the bottom branch (feedback pathway); I think it is much more natural in the top branch of the feedback loop; this may seem counterintuitive, but the error signal is a combined measure of the perception of the reference target (which cannot occur without a delay) minus the perception of self movement (which also has a delay). If you put separate delays on both the reference and the self-movement feedback, each delay occuring "before" the “subtraction bubble”, and assume they are identical delays, then that is equivalent to a single delay after the subtraction bubble. (Note the reference delay and the feedback delay may not be identical, but it in a simplified model, lumping them together is reasonable.) However, the current figure compares an *un-delayed* reference measurement to *delayed* self-movement measurement. Note that writing it this way, with the delay in the top branch (capturing both the delay in feedback and the delay in the measured reference) allows one to lump the sensory delay and the motor delay, since they are all in the top branch and for a linear, single-input—single-output system, the blocks commute in a linear path like this. I don’t think this will change the results (and did not dig carefully enough back into the simulations to see if a change in how the reference jumps and is sensed would matter in the FB controller). It certainly will not improve the FB controller and in fact may make it slightly worse (same stability margins, maximum gains, etc, but perhaps slightly worse performance due to having to wait to sense the change in reference).

2. Reference suggestion: I the sentence “Empirical studies by Horak and colleagues (1996) further showed that human postural responses dynamically reweight sensory feedback sources depending on context and perturbation characteristics [16]” I strongly recommend considering work by Kiemel et al., especially the following:

Kiemel T, Zhang Y, Jeka JJ: Identification of neural feedback for upright stance in humans: stabilization rather than sway minimization. J Neurosci 2011, 31:15144-15153.

3. Line 127, the statement that an FB controller must use lower gains to remain stable is not generally true. Reducing gains is just a common choice for controllers without delay compensation, but it is not a necessary outcome (e.g. with prediction /compensation, different architectures, etc). In other words, “must” is a bit too strong, although it is “typically true”. This could be softened in a reasonable way, e.g. “To compensate for long feedback delays, a feedback-delayed controller often requires lower loop gain above a stability-limiting frequency to remain stable when compared to a controller with shorter delays.”

4. Figure 1 (minor, can ignore if desired), the input torque only reaching saturation in positive limit could be confusing. Perhaps little dashed lines at “Fmax / Fmin” in both the force capacity limits box and in the little inset graphs as this would show that the FB controller just barely stays within the negative limit. Also, it is a bit weird that the FB controller is a deadbeat controller that finishes in finite time, but your paper uses a PD controller (asymptotic).

5. The steady-state torque term is used to compensate for constant gravitational torque at the target posture, and the system will become a pendulum without gravity. But in the manuscript, gravity was just used in scaled model of posture task. To make all these experiments clearer, it’s better to just introduce gravity in scaled model of posture task. For others, the system will become much easier to just give the model without gravity. (Also, the introduction of PID controller is a little confusing, is there really a need to compare a steady state torque term with Integral term?)

6. Line 462: “not reaching negative torque bound” is not an informative conclusion if the reason the negative torque limit is not reached is simply that the initial angle is positive. In other words, if I’m understanding correctly, the asymmetry in +/- clipping is due to the asymmetry in the input. This is fine, but perhaps worth reminding or clarifying (although I’m not sure I’m understanding the significance of this result).

8. Abstract (picky edits that I happened to notice): “full-size range” I don’t think you want the hyphen, it is the “full (size range)” not the “(full-size) range”. Also in the abstract, you don’t need “from literature” since you say in the same sentence it is published (redundant).

Also, I don’t know whether the supporting section S2 (the Bode plot analysis of a linear feedback control system) is needed; it is highly tutorial and this is not a tutorial paper. Also, my student with whom I co-reviewed has a Font error when opening the Supplementary materials on a Mac in Line 80, but it was fine for me; I recommend double-checking a final PDF of the supplementary information on multiple platforms to ensure font compatibility.

**Have the authors made all data and (if applicable) computational code underlying the findings in their manuscript fully available?**

The PLOS Data policy requires authors to make all data and code underlying the findings described in their manuscript fully available without restriction, with rare exception (please refer to the Data Availability Statement in the manuscript PDF file). The data and code should be provided as part of the manuscript or its supporting information, or deposited to a public repository. For example, in addition to summary statistics, the data points behind means, medians and variance measures should be available. If there are restrictions on publicly sharing data or code —e.g. participant privacy or use of data from a third party—those must be specified.requires authors to make all data and code underlying the findings described in their manuscript fully available without restriction, with rare exception (please refer to the Data Availability Statement in the manuscript PDF file). The data and code should be provided as part of the manuscript or its supporting information, or deposited to a public repository. For example, in addition to summary statistics, the data points behind means, medians and variance measures should be available. If there are restrictions on publicly sharing data or code —e.g. participant privacy or use of data from a third party—those must be specified.

Reviewer #1: Yes

Reviewer #2: Yes

PLOS authors have the option to publish the peer review history of their article (what does this mean?). If published, this will include your full peer review and any attached files.). If published, this will include your full peer review and any attached files.

.

Reviewer #1: No

Reviewer #2: No

**Figure resubmission:**
---

## [Decision Letter · Decision Letter 3]

13 Mar 2026

Dear Dr. Mohamed Thangal,

We are pleased to inform you that your manuscript 'Effects of sensorimotor delays and muscle force capacity limits on the performance of feedforward and feedback control in animals of different sizes' has been provisionally accepted for publication in PLOS Computational Biology.

Best regards,

Lyle J. Graham

Section Editor

PLOS Computational Biology

Reviewer's Responses to Questions

**Comments to the Authors:**

Reviewer #2: This is an excellent paper and the authors have addressed all of our concerns.

**Have the authors made all data and (if applicable) computational code underlying the findings in their manuscript fully available?**

The PLOS Data policy requires authors to make all data and code underlying the findings described in their manuscript fully available without restriction, with rare exception (please refer to the Data Availability Statement in the manuscript PDF file). The data and code should be provided as part of the manuscript or its supporting information, or deposited to a public repository. For example, in addition to summary statistics, the data points behind means, medians and variance measures should be available. If there are restrictions on publicly sharing data or code —e.g. participant privacy or use of data from a third party—those must be specified.requires authors to make all data and code underlying the findings described in their manuscript fully available without restriction, with rare exception (please refer to the Data Availability Statement in the manuscript PDF file). The data and code should be provided as part of the manuscript or its supporting information, or deposited to a public repository. For example, in addition to summary statistics, the data points behind means, medians and variance measures should be available. If there are restrictions on publicly sharing data or code —e.g. participant privacy or use of data from a third party—those must be specified.

Reviewer #2: Yes

PLOS authors have the option to publish the peer review history of their article (what does this mean?). If published, this will include your full peer review and any attached files.). If published, this will include your full peer review and any attached files.

.

Reviewer #2: **Yes:**Noah J. CowanNoah J. Cowan

---

## [Editor Report · Acceptance letter]

PCOMPBIOL-D-24-01586R3

Effects of sensorimotor delays and muscle force capacity limits on the performance of feedforward and feedback control in animals of different sizes

Dear Dr Mohamed Thangal,

I am pleased to inform you that your manuscript has been formally accepted for publication in PLOS Computational Biology. Your manuscript is now with our production department and you will be notified of the publication date in due course.

With kind regards,

Anita Estes
